# Downregulation of CD73/A_2A_R-Mediated Adenosine Signaling as a Potential Mechanism of Neuroprotective Effects of Theta-Burst Transcranial Magnetic Stimulation in Acute Experimental Autoimmune Encephalomyelitis

**DOI:** 10.3390/brainsci11060736

**Published:** 2021-06-01

**Authors:** Milorad Dragić, Milica Zeljković, Ivana Stevanović, Marija Adžić, Andjela Stekić, Katarina Mihajlović, Ivana Grković, Nela Ilić, Tihomir V. Ilić, Nadežda Nedeljković, Milica Ninković

**Affiliations:** 1Department for General Physiology and Biophysics, Faculty of Biology, University of Belgrade, 11000 Belgrade, Serbia; milica.zeljkovic@bio.bg.ac.rs (M.Z.); amarija@bio.bg.ac.rs (M.A.); andjela.stekic@bio.bg.ac.rs (A.S.); katarina.mihajlovic@bio.bg.ac.rs (K.M.); nnedel@bio.bg.ac.rs (N.N.); 2Institute for Medical Research, Military Medical Academy, 11000 Belgrade, Serbia; ivanav13@yahoo.ca (I.S.); ninkovic7@gmail.com (M.N.); 3Medical Faculty of Military Medical Academy, University of Defense, 11000 Belgrade, Serbia; tihoilic@gmail.com; 4Department of Molecular Biology and Endocrinology, Vinča Institute of Nuclear Sciences-National Institute of the Republic of Serbia, University of Belgrade, 11000 Belgrade, Serbia; istanojevic@vinca.rs; 5Medical Faculty, University of Belgrade, 11000 Belgrade, Serbia; nelavilic@gmail.com; 6Clinic of Physical Medicine and Rehabilitation, Clinical Center of Serbia, 11000 Belgrade, Serbia

**Keywords:** CD73, adenosine, A_2A_R, A_1_R, neuroinflammation, theta-burst stimulation, rTMS, purinergic signaling

## Abstract

Multiple sclerosis (MS) is a chronic neurodegenerative disease caused by autoimmune-mediated inflammation in the central nervous system. Purinergic signaling is critically involved in MS-associated neuroinflammation and its most widely applied animal model—experimental autoimmune encephalomyelitis (EAE). A promising but poorly understood approach in the treatment of MS is repetitive transcranial magnetic stimulation. In the present study, we aimed to investigate the effect of continuous theta-burst stimulation (CTBS), applied over frontal cranial bone, on the adenosine-mediated signaling system in EAE, particularly on CD73/A_2A_R/A_1_R in the context of neuroinflammatory activation of glial cells. EAE was induced in two-month-old female DA rats and in the disease peak treated with CTBS protocol for ten consecutive days. Lumbosacral spinal cord was analyzed immunohistochemically for adenosine-mediated signaling components and pro- and anti-inflammatory factors. We found downregulated IL-1β and NF- κB-*ir* and upregulated IL-10 pointing towards a reduction in the neuroinflammatory process in EAE animals after CTBS treatment. Furthermore, CTBS attenuated EAE-induced glial eN/CD73 expression and activity, while inducing a shift in A_2A_R expression from glia to neurons, contrary to EAE, where tight coupling of eN/CD73 and A_2A_R on glial cells is observed. Finally, increased glial A_1_R expression following CTBS supports anti-inflammatory adenosine actions and potentially contributes to the overall neuroprotective effect observed in EAE animals after CTBS treatment.

## 1. Introduction

Multiple sclerosis (MS) is a progressive demyelinating and neurodegenerative disorder driven by the adaptive immune response [1,2] and inflicts primary damage to the myelin sheath [3]. Succeeding inflammation and glial cell activation result in diffuse plaques of demyelination and axonal loss in multiple areas of the brain and spinal cord, which are the main cause of progressive neurological disability and motor dysfunctions in MS [4]. The histopathological deteriorations create many other symptoms, including pain, depression, spasticity, and cognitive deficits, which also progress over time [5]. The most common form of MS is relapse-remitting MS (RRMS), occurring in 85% of patients, characterized by symptomatic loss-of-function periods (relapses), followed by complete or partial remissions. Even though a compelling improvement is made regarding the introduction of new disease-modifying treatments, a significant number of patients will still develop a secondary progressive form of MS (SPMS) [6]. Accordingly, new therapeutic, neuroprotective, and myelination supportive approaches able to ameliorate neuroinflammation and neurotoxic reactive phenotype of astrocytes and microglia are major unmet clinical needs in MS [7].

Neuroinflammation driven by astrocytes and microglia in pathological conditions including MS—and its most widely used animal model experimental autoimmune encephalomyelitis (EAE)—is closely regulated by purinergic signaling. Specifically, the neuroinflammatory responses of glial cells begin with an emergence of danger-associated molecular patterns (DAMP), among which adenosine triphosphate (ATP) plays a particular role. Under the pathological conditions, damaged or dying neurons release large amounts of ATP [8,9], which acts at nucleotide-responsive purinoreceptors, P2X or P2Y, to initiate pro-inflammatory actions of glial cells [8]. The action of extracellular ATP at purinoreceptors is ceased by its sequential hydrolysis, mediated by the ectonucleotidase enzyme chain (CD39/NTPDase1, NTPDase2, and CD73). The last step is the hydrolysis of AMP, mediated via ecto-5′-nucleotidase (eN/Cluster of differentiation 73 (CD73)), resulting in the production of adenosine [10,11]. One longitudinal study in MS patients [12] showed that impaired metabolism of extracellular ATP and drop of adenosine in the cerebrospinal fluid were associated with significantly faster disability progression in MS patients over time. Similarly, reduced production of adenosine in blood serum and increased production in the spinal cord tissue [13] were registered along with a strong upregulation of CD73 by reactive astrocytes during the symptomatic phase of EAE [14].

Adenosine, generated by the catalytic action of CD73, acts at P1 receptor subtypes, A_1_, A_2A_, A_2B_, and A_3_, which are abundantly expressed in the CNS [15]. Adenosine plays a critical role in the regulation and complex modal changes in glial cells during neuroinflammation [16]. Although it is generally considered that adenosine, unlike ATP, elicits anti-inflammatory and immunosuppressive effects [17], its effects critically depend on a particular P1 receptor subtype(s), which mediates the adenosine [18]. Thus, concerning MS/EAE, evidence suggests that potentiation of A_1_R and blockade of A_2A_R-mediated adenosine actions induce strong neuroprotective actions via the attenuation of glial cells’ reactivity [15,18,19,20,21,22].

One promising but poorly exploited clinical approach in MS is repetitive transcranial magnetic stimulation (rTMS). rTMS refers to a non-invasive and painless stimulation protocol designed to modulate excitability and activity in several brain systems, by applying magnetic pulses delivered in predefined administration patterns [23,24,25]. Theta-burst stimulation protocol (TBS) is a highly effective version of rTMS, which affords a short stimulation time, low stimulus intensity, and improved reliability of rTMS [26]. Over the past decade, several studies have shown that rTMS stimulation induces measurable clinical outcomes in several neurological disorders, including depression, schizophrenia, stroke, Alzheimer’s disease, and Parkinson’s disease, and significant improvement of motor and cognitive functions in healthy subjects [27,28,29,30,31,32]. So far, studies have demonstrated limited clinical value of rTMS in MS patients [33,34], particularly regarding motor dysfunction [35,36].

Despite numerous positive neurological outcomes in several neurological disorders, mechanisms underlying the plasticity induced by TBS are poorly understood, which urges the need for preclinical animal testing. Up to date, the efficacy of different TBS protocols has been explored in animal studies using EAE, as an experimental paradigm of RRMS. The studies have demonstrated reduced oxidative stress [37,38], attenuation of gliosis [39], and increased expression of brain-derived neurotrophic factor (BDNF) [40]. Therefore, the present study aims to explore the effect of continuous theta-burst stimulation (CTBS) protocol on purinergic system activity in the context of neuroinflammation associated with experimental autoimmune encephalomyelitis in Dark Agouti rats. If proven effective, these data could incite translation into clinical practice as an early/add-on non-invasive therapeutic intervention.

## 2. Material and Methods

### 2.1. Ethical Statement

All experimental procedures were approved by Ethics Comity of Military Medical Academy (Application No. 323-07-00622/2017-05). Care was taken to minimize the pain and discomfort of the experimental animals in accordance with EU Directive 2010/63/EU.

### 2.2. Animals

This study was performed on two months old female Dark Agouti (DA) rats (150–200 g) acquired from Military Medical Academy local colony. All animas were housed under standardized conditions (constant humidity 55 ± 3%, temperature 23 ± 2 oC, 13/11 h light/dark regime) in polyethylene cages (3 animals per cage) with food and water ad libitum.

### 2.3. Induction of Experimental Autoimmune Encephalomyelitis

Acute experimental autoimmune encephalomyelitis was induced as previously described [39]. Briefly, animals were anesthetized with sodium pentobarbital (45 mg/kg, Trittay, Germany) and s.c. injected with 0.1 mL of encephalogenic emulsion comprising complete Freund’s Adjuvant (CFA, 1 mg/mL *Mycobacterium tuberculosis*, Sigma, St. Louis, MO, USA) and rat spinal cord tissue homogenate (50% *w/v* in saline) in right hind foot.

The animals were weighed and daily scored for neurological signs of EAE for 24 days post-injection (dpi) using the standard EAE scoring scale (0–5): 0 = unaffected/no sign of illness; 0.5 = reduced tail tone; 1 = tail atony; 1.5 = slightly clumsy gait, impaired righting ability or combination; 2 = hind limb paresis; 2.5 = partial hind limb paralysis; 3 = complete hind limb paralysis; 3.5 = complete hind limb paralysis accompanied with forelimb weakness; 4 = tetraplegic; 5 = morbidus state or death [41]. Daily score was averaged taking into account all animals within the experimental group.

### 2.4. Theta-Burst Stimulation Protocol

In the present study, theta-burst stimulation (TBS) was applied in the form of continuous protocol (CTBS), as previously described [39,40,42]. Briefly, the stimulation was performed using MagStim Rapid2 device via 25 mm figure-of-eight coil (The MagStim Company, Whitland, UK). Continuous protocol was applied according to [43]. The CTBS block was administered as a single 40 s train of bursts repeated at a frequency of 5 Hz, each block containing 600 pulses. Stimulation intensity was set at 30% of maximal output, just below a motor threshold value. The stimulation was applied by holding the center of the coil directly above the frontal cranial bone in close contact with the scalp of a manually immobilized animal. Given that a coil size is larger than cranium of an animal, application over the frontal cranial bone provides equally distributed whole brain stimulation.

### 2.5. Experimental Groups and Treatment

All animals were divided into four experimental groups: naïve, healthy animals (*n* = 8), EAE animals (sacrificed on day 24, *n* = 8), EAE animals subjected to CTBS protocol (*n* = 8), and animals subjected to sham CTBS noise artifact (*n* = 8). Animals were subjected to either CTBS or noise artifact for 10 consecutive days, starting at 14 dpi, when clinical scoring showed disease peak (Figure 1). The next day, animals were decapitated using Harvard Apparatus, and spinal cord tissue was processed for immunohistochemistry. Given that sham groups did not produce any qualitative/quantitative change when compared to non-treated animals, those images were not shown.

### 2.6. Enzyme Histochemistry

Ectonucleotidase enzyme histochemistry based on the AMP-hydrolyzing activities of and eN/CD73 has been applied, as previously described [44]. Briefly, cryosections were preincubated for 30 min at RT in TRIS-maleate sucrose buffer (TMS), containing 0.25 M sucrose, 50 mM TRIS-maleate, 2 mM MgCl_2_ (pH 7.4), and 2 mM levamisole, to inhibit tissue non-specific alkaline phosphatase. The enzyme reaction was carried out at 37 °C/90 min, in TMS buffer, containing 2 mM Pb(NO_3_)_2_, 5 mM MnCl_2_, 3% dextran T250, and 1 mM substrate (ATP, ADP, or AMP), as substrate. After thorough washing, slides were immersed in 1% (v/v) (NH_4_)_2_S, and the product of enzyme reaction was visualized as an insoluble brown precipitate at a site of the enzyme activity. After dehydration in graded ethanol solutions (70–100% EtOH and 100% xylol), slides were mounted with a DPX-mounting medium (Sigma Aldrich, Saint Louise, MO, USA). The sections were examined under LEITZ DM RB light microscope (Leica Mikroskopie and Systems GmbH, Wetzlar, Germany), equipped with LEICA DFC320 CCD camera (Leica Microsystems Ltd., Heerbrugg, Switzerland) and analyzed using LEICA DFC Twain Software (Leica, Wetzlar, Germany).

### 2.7. Immunofluorescence and Confocal Microscopy

Lumbar areas of the spinal cords (3–4 animals per group) were removed from decapitated animals and fixed in 4% paraformaldehyde (0.1 M PBS, pH 7.4, 12 h at 4 °C) and dehydrated in graded sucrose solution (10–30% in 0.1 M PBS, pH 7.4). After dehydration, 25 µm sections were cut on crytome and collected serially, mounted on supefrost glass slides, air-dried for 1–2 h at room temperature, and stored at 20 °C until staining. After rehydration and washing steps in PBS, sections were blocked with 5% normal donkey serum at room temperature for 1 h, followed by incubation with primary antibodies (Table 1). Slides were then probed with appropriate secondary antibodies (Table 1) for 2 h at room temperature in the dark chamber. Slides were covered using the Mowiol medium (Sigma Aldrich, USA) and left to dry at 4 °C over night. Slides were examined using a confocal laser-scanning microscope (LSM 510, Carl Zeiss, GmbH, Jena, Germany) using Ar multi-line (457, 478, 488, and 514 nm), HeNe (543 nm), HeNe (643 nm) lasers using 63× (×2 digital zoom) DIC oil, 40× and monochrome camera AxioCam ICm1 camera (Carl Zeiss, GmbH, Germany).

### 2.8. Quantification of Immunofluorescence and Multi-Image Colocalization Analysis

All image quantification and analysis were performed using ImageJ software (free download from https://imagej.net/Dowloads, accessed on 10 April 2021). In order to evaluate a degree of overlap and correlation between multiple channels, we performed multi-image colocalization analysis using the JACoP ImageJ plugin. A degree of overlap and correlation between channels was estimated by calculating Pearson’s correlation coefficient (PCC) and Manders’ correlation coefficient (MCC). We captured 7–9 images/animal of the white matter under the same conditions (1024 × 1024, laser gain and exposure) and performed PCC and MCC analysis. Analysis was performed on 40× magnification for PCC and 63× magnifications for MCC analysis. Given that astrocytes and microglia were closely related and often intermingled without clear borders, especially in EAE group, whole images were used for analysis rather than single cell [45]. PCC is a statistical parameter that reflects co-occurrence and correlation of analyzed channels. On the other hand, MCC measures fractional overlap between two signals, signal 1 and signal 2. MCC_1_ quantifies the fraction of signal 1 that co-localizes with signal 2, while MCC_2_ represents the fraction of signal 2 that overlaps with signal 1 [45].

### 2.9. Statistical Analysis

The values are presented either as mean ± SD or SEM, as indicated. Data were first assessed for normality using Shapiro–Wilk followed by adequate parametric test. One-way ANOVA followed by Tuckey post hoc test were used in GraphPad Prism *v*. 6.03. The *p <* 0.05 was considered to be significant (Table 2).

## 3. Results

### 3.1. The Effect of Continuous Theta-Burst Stimulation on the Disease Course

Injection of the encephalitogenic emulsion in susceptible DA rats resulted in a typical acute disease, characterized by gradual neurological deterioration and significant weight loss followed by a spontaneous recovery (Figure 2), as previously reported [39]. Briefly, in the non-treated group (EAE), the first clinical signs of EAE appeared at ~10 post-injection (dpi), peaked at 14 dpi, and withdrew at ~24 dpi. In the group subjected to the CTBS protocol (EAE+CTBS), the stimulation was applied to start from 14 dpi for 10 consecutive days. The effect of the CTBS noise artifact was explored in the sham group of animals (EAE+CTBSpl), which were subjected to the noise artifact according to the same experimental scheme. Significant reduction in duration, disability, and weight loss were observed after CTBS treatment, compared to both sham and naïve animals, as previously published (Figure 2) [37,39].

### 3.2. CTBS Promotes Anti-Inflammatory Milieu in EAE

One of the critical pathological features of EAE/MS is the invasion of peripheral immune cells into the CNS parenchyma and the release of pro-inflammatory mediators, which initiate the neuroinflammatory response of astrocytes and microglia. Therefore, we first examined the effect of CTBS on the inflammatory milieu induced by EAE. IL-1β is a master inflammatory cytokine and the effector molecule in MS/EAE [46]. While control tissue did not express IL-1β-immunoreactive (*ir*) signal (Figure 3A,D), conspicuous IL-1β-*ir*, mostly residing at GFAP-*ir* astrocytes and IBA-1-*ir* microglial cells, were observed in the gray (Figure 3B) and white matter (Figure 3E) of EAE animals, respectively. Prominent IL-1β-*ir* was also observed at neuronal cell bodies in both ventral and dorsal gray matter (Figure 3B). However, the upregulation of IL-1β was completely prevented in EAE animals subjected to CTBS, together with the GFAP-*ir* and Iba-1-*ir* lowered to the level seen in healthy control (Figure 3C,F). The downstream signaling cascade of IL-1β initiates nuclear factor kappa-light-chain-enhancer of activated B cells (NF-κB) family of transcription factors, which trigger the transcription of proinflammatory genes [47]. Strong NF-κB-*ir*, mostly residing at GFAP-*ir* astrocytes cells in EAE animals (Figure 4B, arrowhead), was attenuated to a control level after CTBS treatment protocol (Figure 4C). The cytokine IL-10, on the other hand, exhibits immune response downregulatory properties, which include suppression of the synthesis and release of pro-inflammatory cytokines (PMID: 10320650). Basal IL-10-*ir* in control sections (Figure 5A) was attenuated in EAE (Figure 5B), while CTBS protocol enhanced the intensity of IL-10-*ir* in comparison to control (Figure 5C). The IL-10-*ir* mostly resided at GFAP-*ir* astrocytes (Figure 5C). Sham-treated animals did not show any observable changes when compared to EAE (not shown).

### 3.3. CTBS Attenuates EAE-Induced Expression of CD73

The main objective of the present study was to evaluate the effects of CTBS on purinergic system activity in the context of neuroinflammatory activation of astrocytes and microglia. Hence, we first examined the level of expression and cellular localization of CD73 in the spinal cord tissue in control, non-treated, and CTBS-treated EAE animals (Figure 6). The degree of overlap between CD73 and selected fluorescence signals was determined by calculating PCC and MCC coefficients, which reflect the co-occurrence of selected signals and the fraction of pixels with positive values for selected signals, respectively. In control sections, faint CD73-*ir* was mainly associated with quiescent GFAP-*ir* cells and only sporadically with IBA-1-*ir* microglia (Figure 6A,a). A prominent increase in CD73-*ir* in EAE was mainly associated with IBA-1-*ir* (Figure 6B), which is reflected with the increase in both PCC and MCC_2_ for the two signals, and only marginally with GFAP-*ir* (*p* < 0.05; Figure 6D). The increase in CD73-*ir* was completely reversed by the CTBS treatment (Figure 6C,c), which was reflected with a decrease in MCC_2_ value primarily for CD73-IBA-1, but also for CD73-GFAP overlap (*p* < 0.05, Figure 6E). The occurrence of CD73-*ir* with both fluorescence tracers for astrocytes and microglia was confirmed with the Z-stack imaging (Figure 6F). Interestingly, the fraction of the CD73-*ir* in control and CTBS sections was found without association with GFAP- and IBA-1-*ir* (Figure 6c, arrowheads).

### 3.4. CTBS Attenuates EAE-Induced Upregulation of CD73 and Shift in A_1_R-to-A_2A_R Expression

Altered immunofluorescence imaging directed to CD73 pointed to significant alterations of CD73 expression, both in EAE and after CTBS treatment. Therefore, the expression of the CD73 enzyme activity was shown by AMP-based enzyme histochemistry (Figure 7). The diffuse histochemical reaction produced by CD73-catalyzed hydrolysis of AMP was dominantly observed in the control spinal cord gray matter (Figure 7A,B), whereas the white matter was faintly stained (Figure 7A,C). In EAE sections, an increased reaction was observed in both gray (Figure 7D,E) and white matter (Figure 7D,F), with numerous amoeboid CD73-reactive cells (Figure 7E). Again, CTBS treatment resulted in histochemical staining almost identical to the control (Figure 7G–I). Diffuse staining dominated the ventral and dorsal gray matter (Figure 7G), whereas no infiltrations of amoeboid cells could be found in the white matter (Figure 7H,I).

Signaling actions of adenosine in the CNS are mostly mediated via high-affinity inhibitory A_1_R and excitatory A_2A_R receptors, differentially involved in neuroinflammatory processes [15,18]. In physiological conditions, the expression is dominated by A_1_R mostly found in association with the gray and white matter parenchyma (Figure 8A,a). The induction of EAE is associated with marked loss of A_1_R-*ir*, particularly from the white matter projection pathways (Figure 8B,b). However, CTBS treatment restored and even enhanced the intensity of A_1_R-*ir* (Figure 8C,c). The determination of PCC and MCC had shown that CTBS increases the proportion of both GFAP-*ir* astrocytes and IBA-1-*ir* cells, which expressed A_1_R-*ir*, whereas the overall fraction of A_1_R-*ir* is expressed by the glial cells (Figure 8D,E; *p* < 0.05), also confirmed by Z-stack imaging (Figure 8F). Therefore, EAE is associated with the significant axonal loss of A_1_R-*ir*, whereas CTBS restores the expression and even potentiates it at responsive glial cells.

Concerning the A_2A_R, the intensity of *ir* was weak in control sections, and no significant co-localization was observed with either GFAP-*ir* or IBA-1 (Figure 9A,a). EAE was associated with significant enhancement of A_2A_R-*ir*, particularly co-localized with GFAP- and IBA1*-ir* (Figure 9B,b), reflected through a significant increase in PCC for the association of A_2A_R with GFAP and IBA-1 (Figure 9D). Again, CTBS treatment markedly decreased the intensity of A_2A_R-*ir* and induced massive dissociation between GFAP- and IBA-1-*ir.* A significant part of A_2A_R-*ir* after CTBS resided at 5–7 µm in diameter ovoid structures, probably axon fibers (Figure 9c, arrowhead). Combined immunofluorescence directed to A_2A_R and neurofilament H protein showed a strong association of A_2A_R with neuronal cell bodies in the gray matter and with axonal fibers in the white matter (Figure 10A,B). The CTBS treatment reduced A_2A_R expression on glial cells and increased it on spinal cord neurons.

## 4. Discussion

EAE is a widely used experimental model of the autoimmune neurodegenerative pathology driven by an intertwined network of adaptive immune and CNS resident cells and their inflammatory mediators, which reproduce all the critical events in MS. According to current understanding, pro-inflammatory mediator IL-1β and its main downstream target, NF-κB, are critically involved in the pathogenesis of MS/EAE [48], while the induction of anti-inflammatory cytokine IL-10 correlates with the clinical recovery [49]. The involvement of extracellular ATP, adenosine, and their respective P2 and P1 purinoceptors in the neurodegenerative processes associated with MS/EAE is established as well [50]. Several recent reports emphasize the contribution of ectonucleotidases and ATP/ADP- [41,51,52] and adenosine-mediated signaling in the neuroinflammatory process in EAE pathology (Safarzadeh et al., 2016 [53]; Nedeljkovic, 2019 [18], Lavrnja et al., 2015 [14]; Zhou et al., 2019 [54]). Accordingly, the present study shows that the neuroinflammatory process in EAE is associated with prominent upregulation of CD73 in lumbosacral spinal cord tissue, mostly by reactive microglia and astrocytes activated in response to immune cell invasion to the CNS. Given that CD73 is the only adenosine-producing enzyme in the extracellular milieu [55], the strong induction of CD73 corroborates the finding of the substantial accumulation of adenosine in the extracellular space during EAE (Lavrnja et al., 2009 [13]; 2015 [14]). Although adenosine is generally considered a powerful anti-inflammatory and immunosuppressive molecule [56,57], it exerts pleiotropic actions depending on the functional coupling with particular P1 receptor subtype [15,18,20]. Thus, in physiological conditions, extracellular adenosine, present in low micromolar concentrations, mainly activates inhibitory a A_1_R receptor subtype ubiquitously present in the CNS cell types. However, in neuroinflammatory conditions, the actions of adenosine are mediated largely via excitatory A_2A_R and low-affinity A_2B_R receptor subtypes. Indeed, the upregulation of A_2A_R and its tight spatial coupling with CD73 is another common feature of inflamed tissue in several brain pathologies, including EAE/MS [15,58,59]. Our present study, thus, corroborates the view that the gain-of-function in CD73/A_2A_R and enhanced adenosine signaling drives neuroinflammation and directs the course of EAE.

By using the pathological context of EAE, the principal goal of our study was to show the ability and efficiency of the CTBS protocol to revert the EAE-induced alterations in adenosine signaling and, thus, to point to potential merit of TBS as a therapeutic approach in MS/EAE. Beneficial and anti-inflammatory actions of TBS have been demonstrated in several neurological and psychiatric disorders and animal models, so far [60,61,62,63,64,65]. In the current study, we have observed that animals subjected to CTBS experienced milder neurological dysfunctions for a shorter time than in the group of non-treated EAE. At the histopathological level, the CTBS protocol prevented the release of IL-1β and reduced NF-κB signaling, while increasing the expression of anti-inflammatory IL-10. These effects altogether suggested that CTBS exerted neuroimmune downregulating properties. Indeed, animals subjected to CTBS exhibited significantly lower numbers of reactive microglial cells and hypertrophied astrocytes, which are the typical histological hallmark of the spinal cord tissue injury in EAE [14,39]. The treatment also decreased both the levels of CD73 enzyme activity and the protein expression, particularly by microglia and astrocytes, suggesting a decrease in the extracellular level of adenosine. Given that CD73 itself is necessary for the peripheral T cells entry and the induction of EAE [66,67], altered expression of CD7 by microglia and astrocytes may be seen as the critical factor of the reduced peripheral immune cell entry and local neuroinflammation [18,66].

Besides CD73, the CTBS treatment completely reverted the expression of adenosine receptors, at least the dominant A_1_R and A_2A_R subtypes. Specifically, CTBS prevented the exclusion of A_1_R-mediated signaling observed in EAE and even enhanced the purinoceptors expression in respect to naïve animals. The enhanced expression was mainly observed at astrocytes and microglia, at which the A_1_R receptor activation decreases proinflammatory cytokines and chemokines, thus reducing astrocyte ability to interact with autoreactive CD4^+^ lymphocytes (Liu et al., 2018 [68]; Cunha, 2005 [58]; Liu et al., 2018 [68]; Bijelić et al., 2020 [52]). Furthermore, the CTBS treatment prevented excessive A_2A_R signaling and decreased the co-occurrence of both the A_2A_R and CD73 with the glial cells markers. Instead, CTBS induced neuronal expression of A_2A_R, which is known to regulate the tonic expression and synaptic actions of BDNF [40], thus promoting neuronal survival [69,70,71]. Namely, neuronal A_2A_R-mediated signaling increases BDNF synthesis and the resulting synaptic efficiency and LTD-induced plasticity [72,73], which may be one of the possible mechanisms of the CTBS-induced protective actions in EAE.

In the end, we would like to point out some limitations of our study. Due to size of the TBS stimulation coil, when applied, the whole brain of DA rats is being stimulated, and therefore, we could not ascribe observed beneficial effects to a specific brain region. The beneficial effects observed in this study are most likely mediated via various descending cerebro-spinal tracts. It is possible that focal stimulation of a specific region would yield even better effects; therefore, further research is required in this direction. Another potential limitation would be the selected time of stimulation, since we chose to stimulate animals in the peak of disease and monitor them until the end of disease. Even though it is more common practice to start treatment in the onset of acute EAE, we wanted to examine beneficial effects that could translate to more real situation, since MS patients seek medical attention usually during the peak of their symptoms, which corresponds to the peak of acute EAE in experimental animals.

## 5. Conclusions

Our study convincingly demonstrates that the applied CTBS protocol efficiently counteracts the EAE-induced effects on adenosine signaling and attenuates the reactive state of microglia and astrocytes at histological and biochemical levels, thus providing powerful protective and reparative potential in EAE. Given the paucity of effective treatments in MS, the TBS protocols could be a safe and effective complementary therapeutic approach, together with other disease-modifying treatments, that could provide better clinical outcome in MS.

## Figures and Tables

**Figure 1 brainsci-11-00736-f001:**
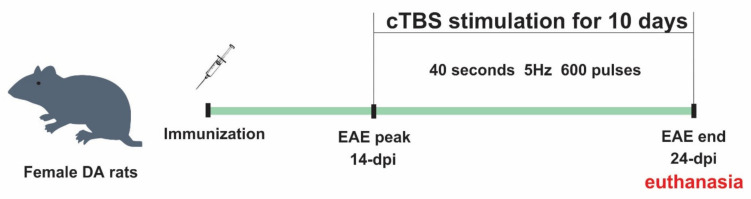
CTBS treatments of EAE rats. Rats were immunized for EAE at day 0 and scored and weighed every day until day 24. The first symptoms appeared around 10 dpi and peaked around 14 dpi. The animals were subjected to CTBS or sham noise artifact for 10 consecutive days from disease peak and euthanized.

**Figure 2 brainsci-11-00736-f002:**
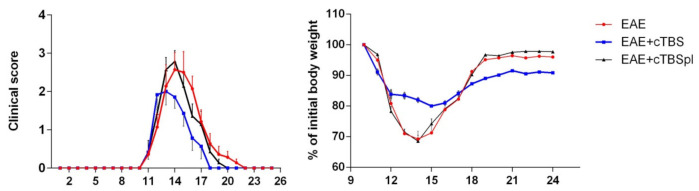
Effects of CTBS treatment on the clinical score of EAE and weight of DA rats. Clinical score and weight of EAE (red circles) in DA rats treated with CTBS protocol (blue square) and CTBS sham noise artifact (black triangles). Animals were monitored from 0 dpi when EAE was induced until 24 dpi when animals were sacrificed.

**Figure 3 brainsci-11-00736-f003:**
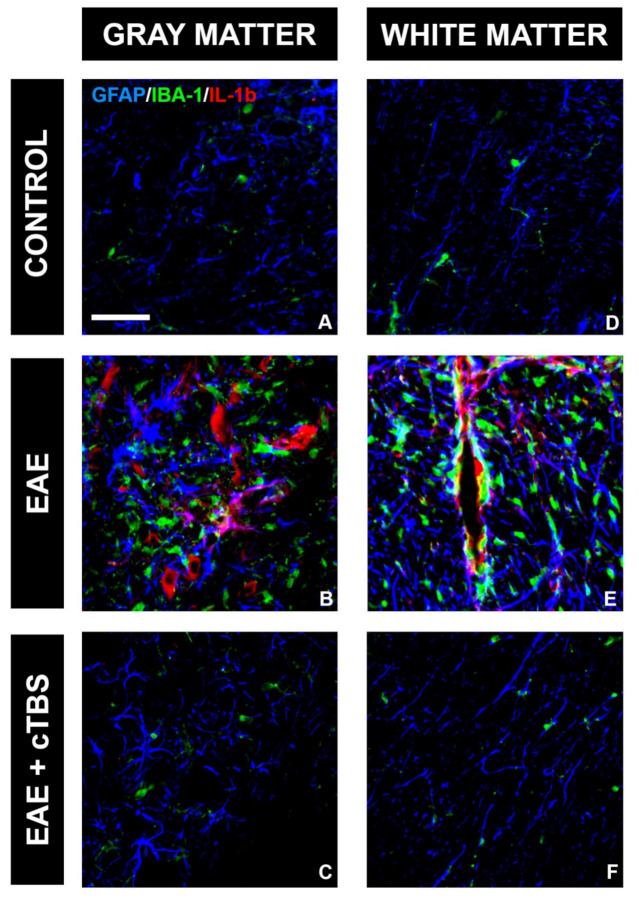
Effect of CTBS treatment on IL-1β expression in gray and white matter of EAE rats. Triple immunofluorescence labeling directed to astrocyte marker GFAP (blue), microglial marker IBA-1 (green), and pro-inflammatory cytokine IL-1β (red). Expression of IL-1β was not detected in control sections (**A**,**D**). In EAE sections, increased IL-1β immunostaining in gray (**B**) and white matter (**E**), colocalizing with both GFAP and IBA-1 cells. After CTBS treatment, no IL-1β-*ir* was observed (**C**,**F**). Scale bar corresponds to 50 μm.

**Figure 4 brainsci-11-00736-f004:**
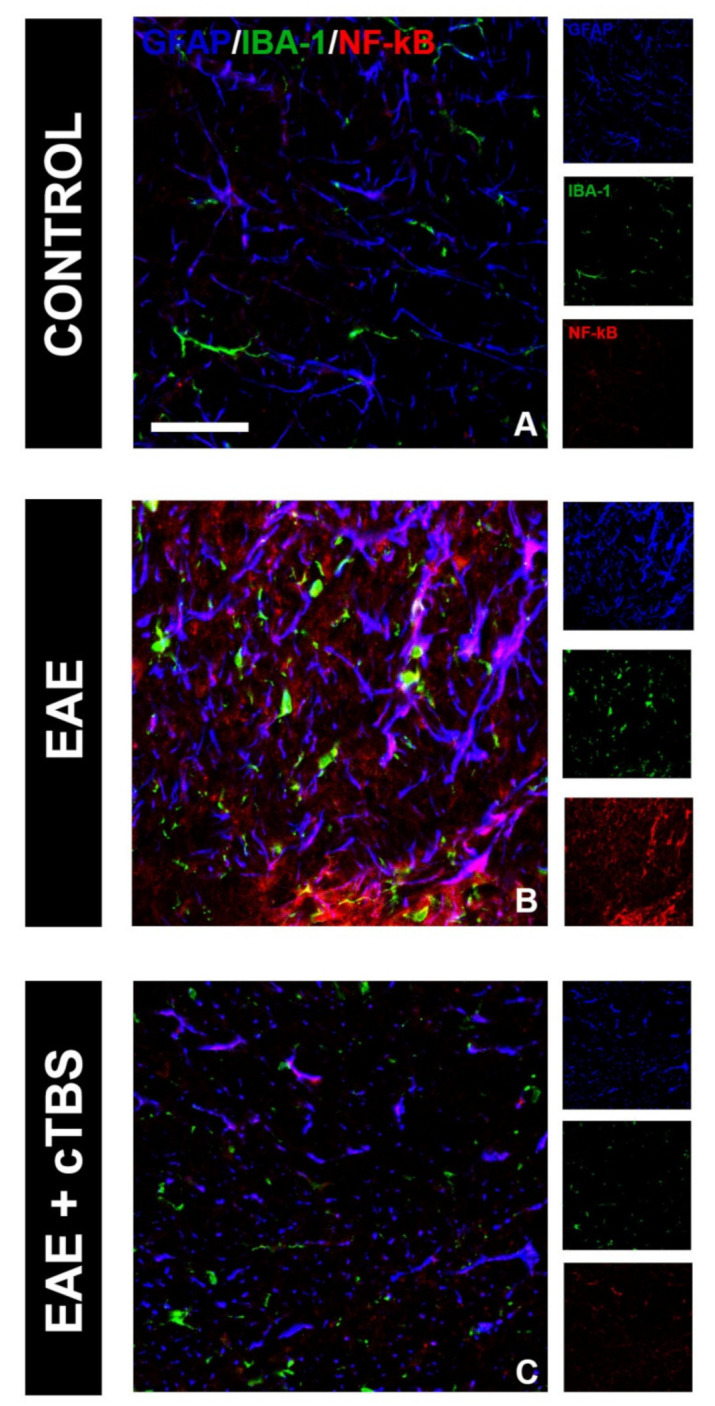
Effects of CTBS treatment on NF-κB expression in EAE rats. Triple immunofluorescence labeling directed to astrocyte marker GFAP (blue), microglial marker IBA-1 (green), and NF-κB (red). Faint colocalization of NF-κB-*ir* and GFAP was observed in control sections (**A**). In EAE sections, a marked increase in NF-κB-*ir* was observed predominantly colocalizing with GFAP^+^ cells (**B**). CTBS treatment decreased immunostaining of NF-κB, and only scattered NF-κB^+^/GFAP^+^ cells were observed (**C**). Scale bar corresponds to 50 μm.

**Figure 5 brainsci-11-00736-f005:**
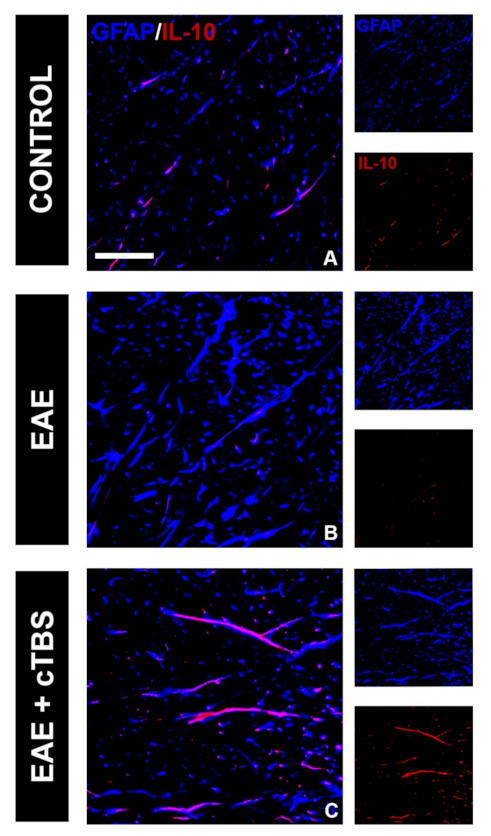
Effects of CTBS treatment on IL-10 expression in EAE rats. Double immunofluorescence labeling directed to astrocyte marker GFAP (blue) and anti-inflammatory cytokine IL-10 (red). Control sections revealed modest colocalization of IL-10 and GFAP (**A**), which was barely detectable in EAE animas (**B**). CTBS treatment led to marked increase in immunostaining of IL-10, which was confined to quiescent GFAP^+^ cells (**C**). Scale bar corresponds to 50 μm.

**Figure 6 brainsci-11-00736-f006:**
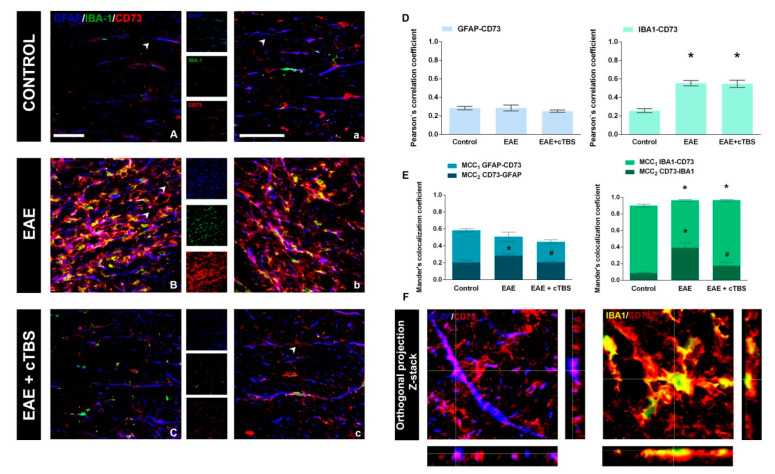
Effects of CTBS treatment on eN/CD73 expression in EAE rats. Triple immunofluorescence labeling directed to astrocyte marker GFAP (blue), microglial marker IBA-1 (green), and eN/CD73 (red). In control section, faint staining of eN/CD73 was observed colocalizing dominantly with GFAP^+^ cells (**A,a**). In EAE sections, a marked increase in eN/CD73 staining was observed colocalizing with GFAP^+^ and IBA-1^+^ cells (**B,b**). After CTBS treatment, a significant reduction in eN/CD73-*ir* was observed (**C,c**). Pearson correlation coefficients (PCC) indicating the level of signal overlap between GFAP-*ir* and eN/CD73-*ir* and IBA-1-*ir* and eN/CD73-*ir*. Bars show mean PCC ± SEM, from 7–9 images/animal (**D**). Mander’s colocalization coefficient (MCC) indicating level of signal colocalization between GFAP/CD73 (MCC_1_, light blue), CD73/GFAP (MCC_2_, dark blue), IBA-1/CD73 (MCC_1_, light green), and CD73/IBA1 (MCC_2_, dark green) (**E**). Orthogonal Z-stack projection of GFAP/CD73 and IBA-1/CD73 (**F**). Level of significance: * *p* ˂ 0.05 or less when compared to control, ^#^
*p* ˂ 0.05 when compared to EAE. Scale bar corresponds to 50 μm.

**Figure 7 brainsci-11-00736-f007:**
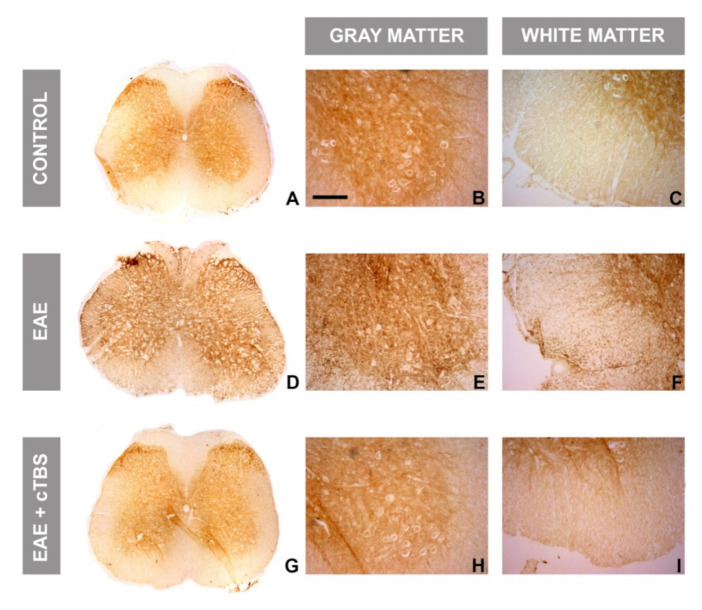
Effects of CTBS treatment on AMP-based enzyme histochemistry in lumbar spinal cords of EAE rats. Enzyme histochemistry in the presence of AMP as a substrate labeling structures that exhibit eN/CD73 activity in the spinal cord of control, EAE, and CTBS-treated EAE sections. Control sections (**A**) exhibited diffuse staining patterns localized mainly in gray matter (**B**), while white matter was devoid of staining. (**C**) EAE sections reveled (**D**) a marked increase in eN/CD73 activity localized in gray (**E**) and white matter (**F**). After CTBS protocol (**G**), faint activity was observed in both gray (**H**) and white matter (**I**), similarly to control sections. Scale bar corresponds to 50 μm.

**Figure 8 brainsci-11-00736-f008:**
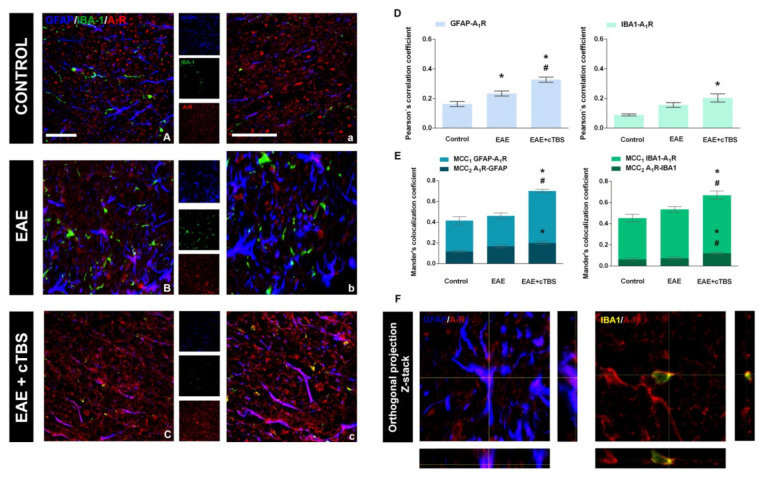
Effects of CTBS treatment on A_1_R expression in lumbar spinal cords of EAE rats. Triple immunofluorescence labeling directed to astrocyte marker GFAP (blue), microglial marker IBA-1 (green), and A_2A_R (red). In control sections, moderate staining of A_1_R*-ir* was observed mostly confined to what appeared to be neuronal elements (**A,a**). In EAE sections, no apparent change in A_1_R*-ir* was observed compared to control (**B,b**). After CTBS treatment A_1_R*-ir* was significantly increased on glial cells (**C,c**). Pearson correlation coefficients (PCC) indicating the level of signal overlap between GFAP-*ir* and A_1_R-*ir* and IBA-1-*ir* and A_1_R-*ir*. Bars show mean PCC ± SEM, from 7–9 images/animal (**D**). Mander’s colocalization coefficient (MCC) indicating level of signal colocalization between GFAP/A_1_R *(*MCC_1_, light blue), A_1_R/GFAP (MCC_2_, dark blue), IBA-1/A_1_R (MCC_1_, light green), and A_1_R/IBA1 (MCC_2_, dark green). Bars show mean MCC ± SEM, from 7–9 images/animal (**E**). Orthogonal Z-stack projection of GFAP/A_1_R and IBA-1/A_1_R (**F**). Level of significance: * *p* ˂ 0.05 or less when compared to control, ^#^
*p* ˂ 0.05 when compared to EAE. Scale bar corresponds to 50 μm.

**Figure 9 brainsci-11-00736-f009:**
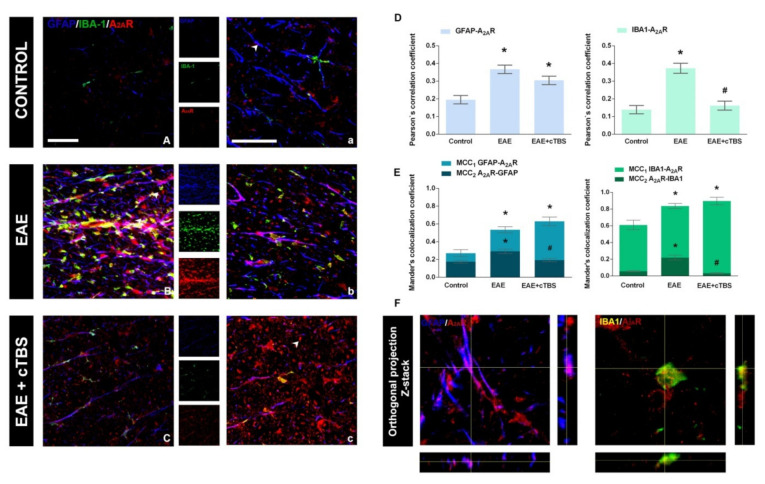
Effects of CTBS treatment on A_2A_R expression in lumbar spinal cords of EAE rats. Triple immunofluorescence labeling directed to astrocyte marker GFAP (blue), microglial marker IBA-1 (green), and A_2A_R (red). In control sections, faint staining of A_2A_R*-ir* was observed (**A,a**). A prominent increase in A_2A_R*-ir* was observed in EAE, both in association with GFAP- and IBA-1-*ir* (**B,b**). The CTBS treatment decreased the overall intensity of A_2A_R*-ir* in the gray matter and was reduced on glial cells, but an increase in staining was detected in non-glial elements (**C****,c**). Pearson correlation coefficients (PCC) indicating the level of signal overlap between GFAP-*ir* and A_2A_R-*ir* and IBA-1-*ir* and A_2A_R-*ir*. Bars show mean PCC ± SEM, from 7–9 images/animal (**D**). Mander’s colocalization coefficient (MCC) indicating level of signal colocalization between GFAP/A_2A_R (MCC_1_, light blue), A_2A_R/GFAP (MCC_2_, dark blue), IBA-1/A_2A_R (MCC_1_, light green), and A_2A_R/IBA1 (MCC_2,_ dark green) Bars show mean MCC ± SEM, from 7–9 images/animal (**E**). Orthogonal Z-stack projection of GFAP/A_2A_R and IBA-1/A_2A_R (**F**). Level of significance: * *p* ˂ 0.05 or less when compared to control, ^#^
*p* ˂ 0.05 when compared to EAE. Scale bar corresponds to 50 μm.

**Figure 10 brainsci-11-00736-f010:**
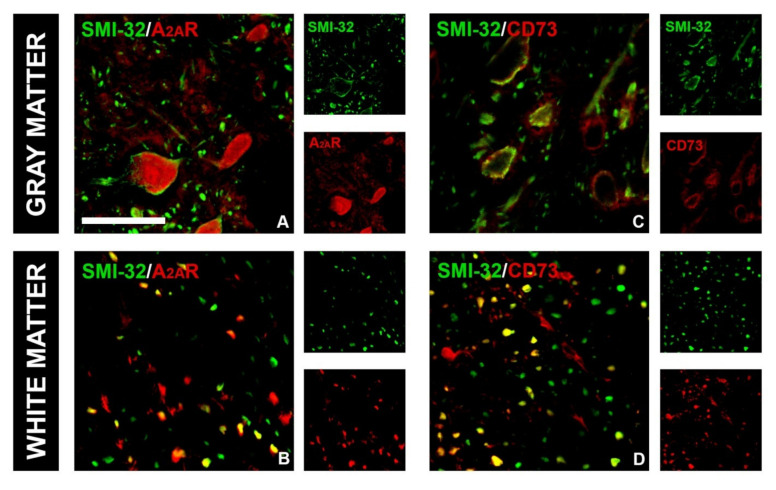
A_2A_R and CD73 expression in gray and white matter in lumbar spinal cords of CTBS-treated rats. A_2A_R signal was colocalized with SMI-32: in gray matter, co-staining was observed in neuronal soma (**A**), whereas in white matter, neuronal axons showed A_2A_R immunoreactivity (**B**). CD73 signal was colocalized with SMI-32: in gray matter (**C**), whereas in white matter, neuronal axons showed SMI-32/CD73 colocalization (**D**). Scale bar corresponds to 50 μm.

**Table 1 brainsci-11-00736-t001:** Antibodies used for immunohistochemistry.

Antibody	Source and Type	Used Dilution	Manufacturer
Iba-1	Goat, polyclonal	1:400	Abcam ab5076, RRID:AB_2224402
CD73, rNu-9L(I4,I5)	Rabbit, polyclonal	1:300	Ectonucleotidases-ab.com
GFAP	Rabbit, polyclonal	1:500	DAKO, Agilent Z0334, RRID:AB_10013382
IL-10	Goat, polyclonal	1:100	Santa Cruz Biotechnology, sc-1783, RRID: AB_2125115
NF-kB	Rabbit, polyclonal	1:100	Santa Cruz Biotechnology, sc-109, RRID: AB_632039
IL-1β/IL-1F2	Goat, polyclonal	1:100	R&D Systems, AF-501-NA, RRID: AB_ 354508
A2AR	Rabbit, polyclonal	1:300	Abcam, ab3461, RRID: AB_303823
A1R	Rabbit, polyclonal	1:200	Novus Biologicals, NB300-549, RRID: AB_10002337
Anti-mouse IgG Alexa Fluor 488	Donkey, polyclonal	1:400	Invitrogen A21202, RRID:AB_141607
Anti-goat IgG Alexa Fluor 488	Donkey, polyclonal	1:400	Invitrogen A-11055, RRID:AB_142672
Anti-rabbit IgG Alexa Fluor 555	Donkey, polyclonal	1:400	Invitrogen A-21428, RRID:AB_141784
Anti-mouse IgG Alexa Fluor 647	Donkey, polyclonal	1:400	Thermo Fisher Scientific A-31571, RRID:AB_162542

**Table 2 brainsci-11-00736-t002:** Results of ANOVA analysis performed for results obtained from image analysis.

Analysis Performed	ANOVA Results	*p* Values
PCC GFAP–CD73	F_(2, 28)_ = 0.7792	*p =* 0.4736
PCC IBA1–CD73	F_(2, 31)_ = 33.48	*p <* 0.0001
MCC1 GFAP–CD73	F_(2, 28)_ = 3.228	*p =* 0.0648
MCC2 CD73-GFAP	F_(2, 28)_ = 4.975	*p <* 0.05
MCC1 IBA1–CD73	F_(2, 27)_ = 5.482	*p <* 0.05
MCC2 CD73–IBA1	F_(2, 27)_ = 17.05	*p <* 0.0001
PCC A_1_R–CD73	F_(2, 30)_ = 22.19	*p <* 0.0001
PCC IBA1–A_1_R	F_(2, 28)_ = 9.155	*p <* 0.01
MCC1 GFAP–A_1_R	F_(2, 27)_ = 24.45	*p <* 0.0001
MCC2 A_1_R-GFAP	F_(2, 27)_ = 9.217	*p <* 0.01
MCC1 IBA1–A_1_R	F_(2, 28)_ = 9.502	*p <* 0.01
MCC2 A_1_R–IBA1	F_(2, 28)_ = 5.458	*p <* 0.05
PCC GFAP–A_2A_R	F_(2, 33)_ = 12.74	*p <* 0.001
PCC IBA1–A_2A_R	F_(2, 32)_ = 25.23	*p <* 0.0001
MCC1 GFAP–A_2A_R	F_(2, 26)_ = 20.86	*p <* 0.0001
MCC2 A_2A_R-GFAP	F_(2, 27)_ = 8.629	*p <* 0.01
MCC1 IBA1–A_2A_R	F_(2, 29)_ = 10.93	*p <* 0.001
MCC2 A_2A_R–IBA1	F_(2, 29)_ = 31.472	*p <* 0.0001

## Data Availability

The datasets generated during and/or analyzed during the current study are available from the corresponding author on reasonable request.

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
