# Peer review of "Downregulation of CD73/A2AR-Mediated Adenosine Signaling as a Potential Mechanism of Neuroprotective Effects of Theta-Burst Transcranial Magnetic Stimulation in Acute Experimental Autoimmune Encephalomyelitis"

_brainsci, 2021, doi:10.3390/brainsci11060736_

Round 1

Reviewer 1 Report

The manuscript presents an interesting experiment with potential clinical applicability. The paper is very well written and the rational for testing the effectiveness of theta burst stimulation to improve the clinical outcome and to reduce some of  the neuroinflammatory reactions associated to MS is clearly stated.

The design is also sound and the selection of control groups is appropriate.

I have, though, several concerns:

  1. In the introduction and methods the reasons why cTBS is applied over the frontal bone should be explained
  2. The location of cTBS should be indicated also in the abstract
  3. In the discussion section a statement about possible mechanisms whereby frontal cortex cTBS may impact on lumbar spinal cord neuroinflammatory changes should be added.
  4. If EAE is an experimental paradigm of RRMS, the aim should emphasize that early cTBS may help reduce the level of inflammation and the duration of rellapses and eventually may help reduce the probability of developing the progressive form (of course, the latter assumption should be tested by means of long-term studies and is out of the scope of the present work). 
  5. Introduction: while it is clear and well written, I suggest to change one statement in lines 49-50. Specifically, the proportion of RRMS patients that eventually develop a progressive form has been reduced in the last years, thanks to novel and early interventions. Could early cTBS be one such interventions that can ameliorate the symptoms of RRMS relapses and reduce the probability of developing the progressive form?
  6. The most important concern is related to the results section. The authors present both qualitative and quantitative data. They state that ANOVA, followed if needed by post-hoc tuckey comparisons between pairs of groups, was used for the analyses of the quantitative data. However, when they describe quantitative differences between pairs of groups (in the clinical symptoms and in the immunohistochemical data), only marginal statistics are shown. The authors should include all the significant ANOVA statistics (F, degrees of freedom and P values), as well as the P of the significant Tuckey post-hoc comparisons.
  7. I suggest adding survival analysis to determine whether the duration of the clinical signs of the different groups differed.
  8. The captions of immunohistochemistry figures should specify the location (lumbar spinal cord).
  9. Did the authors examine/find differences between the right and left sides of the spinal cord?

Author Response

Authors thank the Reviewer for his comments which might improve the quality of our paper. Reviewer comments are in black while Author's responses are in red.

1. In the introduction and methods the reasons why cTBS is applied over the frontal bone should be explained.

We would like to thank the Reviewer for his comment. We have added appropriate explanation in the Material & Methods section. Furthermore, we have added at the end of discussion the part about limitations of the study. 

2. The location of cTBS should be indicated also in the abstract

Thank you for your comment. We have added it in the abstract.

3. In the discussion section a statement about possible mechanisms whereby frontal cortex cTBS may impact on lumbar spinal cord neuroinflammatory changes should be added.

Authors thank the Reviewer for his suggestion. We have added a comment in the limitation study part at the end of the discussion. 

4. If EAE is an experimental paradigm of RRMS, the aim should emphasize that early cTBS may help reduce the level of inflammation and the duration of rellapses and eventually may help reduce the probability of developing the progressive form (of course, the latter assumption should be tested by means of long-term studies and is out of the scope of the present work).

Thank you for your comment. We have added an appropriate sentence at the end of introduction, in the aim part.

5. Introduction: while it is clear and well written, I suggest to change one statement in lines 49-50. Specifically, the proportion of RRMS patients that eventually develop a progressive form has been reduced in the last years, thanks to novel and early interventions. Could early cTBS be one such interventions that can ameliorate the symptoms of RRMS relapses and reduce the probability of developing the progressive form?

Thank you for your suggestion. We have reviewed this part and changed in according to your suggestions. 

6. The most important concern is related to the results section. The authors present both qualitative and quantitative data. They state that ANOVA, followed if needed by post-hoc tuckey comparisons between pairs of groups, was used for the analyses of the quantitative data. However, when they describe quantitative differences between pairs of groups (in the clinical symptoms and in the immunohistochemical data), only marginal statistics are shown. The authors should include all the significant ANOVA statistics (F, degrees of freedom and P values), as well as the P of the significant Tuckey post-hoc comparisons.

Thank you for your comment. Given that there is quite a number of ANOVA analysis performed, we have summarized the results of ANOVA in a Table 2, while post hoc test could be seen in the Figure legends and in Results section.

7. I suggest adding survival analysis to determine whether the duration of the clinical signs of the different groups differed.

Authors thank the Reviewer for his comment. We have previously published those data and we have cited it appropriately in the Results section, subsection 3.1.

8. The captions of immunohistochemistry figures should specify the location (lumbar spinal cord).

Thank you for your comment. We have added specific location to captions of figures.

9. Did the authors examine/find differences between the right and left sides of the spinal cord?

Thank you for your question. We have examined whole sections but did not find any differences between left/right side. 

Reviewer 2 Report

 I suggest you to shorten the introduction and put more recent references. 

I think that the conclusion "Given the paucity of effective treatments in MS, the TBS protocols could be a safe and effective therapeutic approach that could provide better clinical outcome in MS" must be reformulated: cTBS protocol could be a complementary treatment, together with the other disease-modifying treatments.  

Author Response

Authors thank the Reviewer for his comments which might improve the quality of our paper. Reviewer comments are in black while Author's responses are in red.

  1. I suggest you to shorten the introduction and put more recent referencesThank you for your suggestion. We have shortened the introduction and added several more recent references to each section disseminated in the introduction.
  2. I think that the conclusion "Given the paucity of effective treatments in MS, the TBS protocols could be a safe and effective therapeutic approach that could provide better clinical outcome in MS" must be reformulated: cTBS protocol could be a complementary treatment, together with the other disease-modifying treatments. Thank you for your suggestion. We have added the paragraph Reviewer suggested and it now states:

    ``...Given the paucity of effective treatments in MS, the TBS protocols, , could be a safe and effective complementary therapeutic approach, together with other disease-modifying treatments, that could provide better clinical outcome in MS.``

Round 2

Reviewer 1 Report

The authors have adequately addressed the issues of the former version.